# Dense and accurate whole-chromosome haplotyping of individual genomes

David Porubsky[1,8], Shilpa Garg[2,3,4], Ashley D. Sanders[5,6], Jan O. Korbel [5], Victor Guryev[1], Peter M. Lansdorp[1,6,7] & Tobias Marschall [2,3]

The diploid nature of the human genome is neglected in many analyses done today, where a genome is perceived as a set of unphased variants with respect to a reference genome. This lack of haplotype-level analyses can be explained by a lack of methods that can produce dense and accurate chromosome-length haplotypes at reasonable costs. Here we introduce an integrative phasing strategy that combines global, but sparse haplotypes obtained from strand-specific single-cell sequencing (Strand-seq) with dense, yet local, haplotype information available through long-read or linked-read sequencing. We provide comprehensive guidance on the required sequencing depths and reliably assign more than 95% of alleles (NA12878) to their parental haplotypes using as few as 10 Strand-seq libraries in combination with 10-fold coverage PacBio data or, alternatively, 10X Genomics linked-read sequencing data. We conclude that the combination of Strand-seq with different technologies represents an attractive solution to chart the genetic variation of diploid genomes.

[1] European Research Institute for the Biology of Ageing, University Medical Center Groningen, University of Groningen, Building 3226, 9713 AV Groningen, The Netherlands. [2] Center for Bioinformatics, Saarland University, Saarland Informatics Campus E2.1, 66123 Saarbrücken, Germany. [3] Max Planck Institute for Informatics, Saarland Informatics Campus E1.4, 66123 Saarbrücken, Germany. [4] Graduate School of Computer Science, Saarland University, Saarland Informatics Campus E1.3, 66123 Saarbrücken, Germany. [5] European Molecular Biology Laboratory (EMBL), Genome Biology Unit, Meyerhofstraße 1, 69117 Heidelberg, Germany. [6] Terry Fox Laboratory, BC Cancer Agency, 601 West 10th Avenue, Vancouver, BC V5Z 1L3, Canada. [7] Department of Medical Genetics, University of British Columbia, 2350 Health Science Mall, Vancouver, BC V6T 1Z3, Canada. [8] Present address: Max Planck Institute for Informatics, Saarbrücken, Germany. David Porubsky, Shilpa Garg and Ashley D. Sanders contributed equally to this work. Correspondence and requests for materials should be addressed to T.M. (email: t.marschall@mpi-inf.mpg.de)

Human genomes are diploid and possess two copies of each chromosome—one paternal and one maternal copy. At the DNA sequence level, these two homologous copies differ at a number of loci along each chromosome. Such heterozygous variants include single nucleotide variants (SNVs), short indels, as well as larger structural variants such as deletions, duplications, or inversions that change the copy number or orientation of segments of the genome. Discriminating and phasing alleles to their respective parental homolog is valuable in many areas of human genetics. For instance, resolving haplotype structure is required to track inheritance in human pedigrees and populations[1], map regions of meiotic recombination[2,3], identify variant-disease associations[4], detect instances of compound heterozygosity, and study allele-specific events like DNA methylation or gene expression[5]. In particular, long-range haplotype information is needed to systematically study epistatic interactions between variants in enhancers and variants in their target genes or their promotors. This is critical as many variants that have been linked to traits in genome-wide association studies reside in (super) enhancers[6] and enhancer-specific variants can show epistatic effects among one another[7], as well as with their target genes that are beyond the reach of linkage disequilibrium[8]. To better understand these epistatic interactions, we must move beyond merely locating variant alleles and additionally study their functional relationships over long distances. Constructing genome-wide chromosome-length haplotypes is therefore the clear next step to build a more complete picture of genome architecture and function.

Currently, methods used to chart the unique variation of individual human genomes rely largely on second- and third-generation DNA sequencing and can include specialized experimental protocols[9–13]. Sequencing technologies sample the human genome in the form of relatively short molecules (reads) and every read that spans at least two heterozygous variants can essentially be considered as a "mini haplotype" that can be assembled into longer haplotype segments by partially overlapping reads spanning the same variable locus[4]. To this end, haplotype-informative reads need to be partitioned into two disjoint sets that represent the two haplotypes. This process, however, is complicated by errors in sequencing as well as genotyping. For these reasons, assembling haplotypes directly from sequencing data is computationally challenging, and the resulting optimization problems are provenly hard[14,15]. Notwithstanding, a number of computational approaches for read-based phasing have recently been developed[16] and, particularly, progress on fixed-parameter tractable algorithms has enabled solving read-based phasing in practice[17–19], for instance through the implementations available in the software package WhatsHap[20]. Beyond phasing reads aligned to a reference genome, various approaches for haplotype-resolved de novo assembly have been explored[21–25].

However, all approaches to reconstruct haplotypes from sequencing reads, be it reference-based or reference-free, come with the intrinsic limitation that the distance between subsequent heterozygous markers can be larger than the read length itself. While long-read sequencing (such as PacBio SMRT[26] and Oxford NanoPore MinION[27]), or linked-read data (such as those provided by 10X Genomics[28]) help to mitigate this issue, these technologies fail to phase over longer stretches of homozygosity, repeat-rich areas including segmental duplications, and centromeres. Thus, specialized techniques that enable homologous chromosomes to be discriminated are required to physically

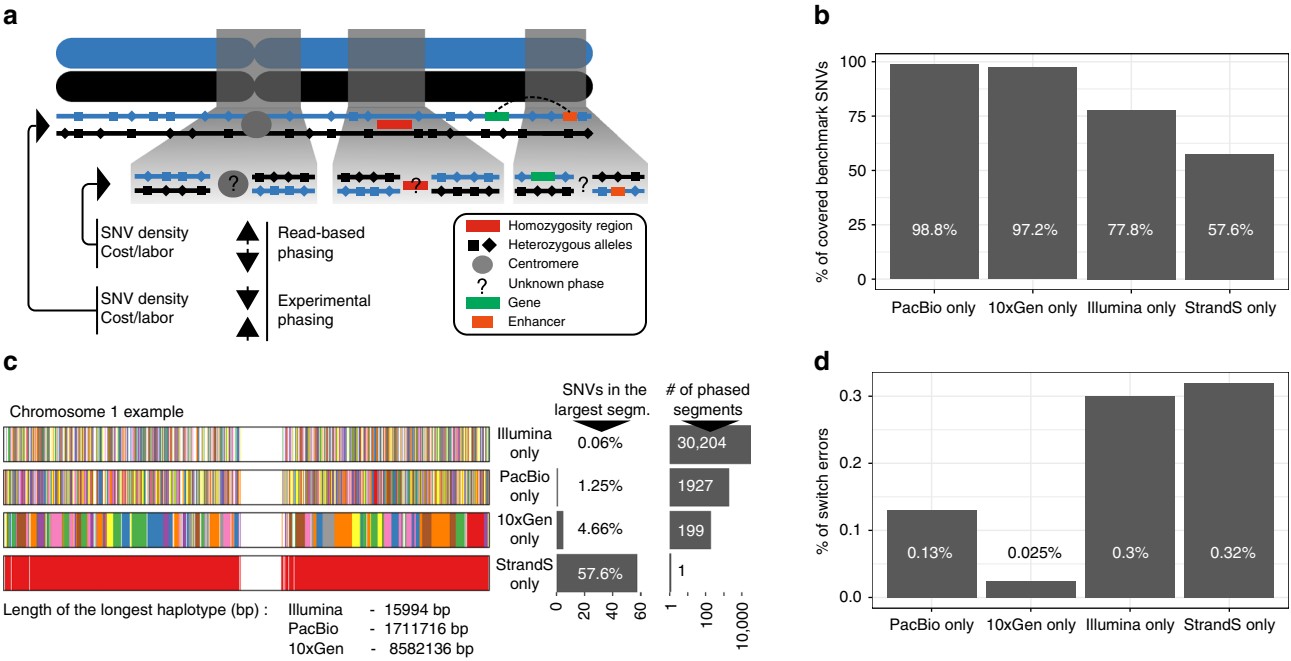

**Fig. 1** Phasing efficacy of read-based and experimental phasing approaches using chromosome 1 as an example. **a** Two homologous chromosomes are shown (blue and black). Experimental phasing approaches like Strand-seq can connect heterozygous alleles along whole chromosomes, however, at higher costs (time and labor) and lower density of captured alleles. In contrast, read-based phasing can deliver high-density haplotypes, but only short haplotype segments are assembled with an unknown phase between them. **b** Barplot showing the percentage of phased variants, for each sequencing technology, from the total number of reference SNVs (Illumina platinum haplotypes). **c** Graphical summary of phased haplotype segments for Illumina, PacBio, 10X Genomics, and Strand-seq phasing shown for chromosome 1. Each haplotype segment is colored in a different color with the longest haplotype colored in red. Side bar graph reports the percentage of SNVs phased in the longest haplotype segment. **d** Accuracy of each independent phasing approach measured as the percentage of switch errors in comparison to benchmark haplotypes

connect alleles across whole chromosomes[3,29,30]. As an alternative to whole-chromosome separation, chromatin capture (Hi-C) methods[31] can be employed to infer long-range haplotype information, based on the assumption that a chromosome will be cross-linked to itself more often than to its homolog[13]. Recently, Hi-C data have been used in combination with other sequencing methods for long-range phasing[32,33]. However, it has been shown that to generate a reliable long-range haplotype scaffold, relatively high sequence coverage (ideally ~90-fold) is needed to reduce bias caused by crosslinks between non-homologous chromosomes[32]. In particular, because these haplotypes need to be inferred statistically, the probability that two heterozygous variants are correctly phased relative to each other deteriorates with increasing chromosomal distances.

Here, we introduce a strategy to obtain dense and global haplotypes that span centromeres, homozygosity regions, and genome assembly gaps, while keeping error rates, costs, and labor at minimum. To this end, we harness the long-range phasing information provided by single-cell template strand sequencing (Strand-seq)[34,35]. Strand-seq is an effective method to assemble highly accurate chromosome-length haplotypes, albeit with lower density of phased alleles in comparison to read-based phasing[9]. Unlike other haplotyping methods, Strand-seq by design distinguishes parental homologs based on the directionality of single-stranded DNA. Therefore, Strand-seq is able to deliver global haplotypes, and its capability to correctly phase two variants with respect to each other does not depend on their distance. To fully exploit this advantage, while at the same time generating dense haplotypes that contain virtually all heterozygous SNVs, we designed a novel unified statistical framework to combine Strand-seq data with short-read, long-read, or linked-read sequencing data. Previously, Strand-seq data were used for phasing on its own, resulting in global yet sparse haplotypes[9]. We demonstrate how the long-range phase information inherent to Strand-seq data can be leveraged to bridge phased segments obtained from Illumina, PacBio, or 10X Genomics sequencing data into contiguous and global haplotypes that span whole chromosomes. We further offer extensive experimental guidance on favorable combinations of the number of used Strand-seq libraries and the depth of PacBio or Illumina coverage, and thus enable considerable reductions in costs and labor—yielding a novel, affordable, and scalable approach for reconstruction of haplotype-resolved individual genomes.

## Results

**Experimental design and data set description**. To explore a new integrative phasing strategy, with the aim of obtaining dense and accurate chromosome-length haplotypes, we used sequencing data available for a well-studied individual (NA12878). The NA12878 genome has been extensively sequenced using multiple technologies, providing high-coverage public sources of sequence information (see "Methods" section). In this study, we focused on read-based phasing data generated from Illumina short-read sequencing and PacBio technology, as they represent current standards for short- and long-read sequencing, respectively (Illumina short-read sequencing is for simplicity referred to as "Illumina data"). The Illumina data set was sequenced to an average depth of 41.1× coverage with a median insert size of 433 bp, and the PacBio data set was sequenced to 45.8× coverage with an average read length of ~4.4 kb (Supplementary Table 1). In addition, we evaluated the performance of 10X Genomics, an emergent linked-read technology. Since none of these technologies alone provides chromosome-length haplotype information, we additionally incorporated single-cell Strand-seq data[9], which has the capacity to scaffold haplotype information obtained from

other data types (Fig. 1a). Here, we used 134 single-cell libraries sequenced to an average depth of 0.037× coverage per library using a paired-end sequencing protocol (see "Methods" section and Supplementary Table 2). To evaluate the phasing accuracy of haplotypes reported in this study, we used the publicly available Illumina platinum haplotypes generated for the same individual (NA12878) as a "reference" standard (see "Methods" section). NA12878 "reference haplotypes" were completed by genetic haplotyping using highly accurate genotypes from 17 individuals of a three-generation pedigree[36], which renders it an ideal gold-standard set for haplotype comparisons. We confirmed that sites and genotypes are in very good agreement with Genome in a Bottle calls (Supplementary Note 1). However, it should be noted that, due to stemming from short reads, this SNV set most likely lacks some variants at repetitive or complex genomic loci (e.g., recent segmental duplications).

**Phasing performance of individual technologies**. To independently assess the phasing performance of each technology, we assembled haplotypes directly from sequencing reads (Illumina or PacBio) using WhatsHap (see "Methods" section). The main advantage of this algorithm is that it solves the minimum error correction (MEC) problem optimally with a runtime that scales linearly in the number of variants (alleles) and is independent of the read length. Therefore, it performs well with short-read technologies (Illumina) and is especially suited for use with long reads (PacBio, Oxford NanoPore). 10X Genomics haplotype segments were assembled by the vendor using the 10X Long-Ranger pipeline. To phase multiple Strand-seq libraries, we have developed a new phasing algorithm, implemented in the R package StrandPhaseR (see "Methods" section and Supplementary Fig. 1). In comparison to our previously published phasing algorithm[9], the current algorithm is provided as an easy to use R package and implements a more robust heuristic approach to solve the MEC problem for Strand-seq data. The haplotypes generated by each technology (i.e., Illumina, PacBio, 10X Genomics, and Strand-seq) were compared to the Illumina platinum reference haplotypes to establish the density, completeness, and accuracy of the phased blocks delivered by each platform independently. For a more streamlined exposition, we focus on results obtained for chromosome 1 in the following analysis and present numbers aggregated across all chromosomes in a concluding discussion.

We found both PacBio and 10X Genomics technologies capable to phase nearly the complete set of variants listed in the reference haplotypes (98.8 and 97.2%, respectively), whereas Illumina alone phased only 77.8% and Strand-seq only 57.6% of the reference SNVs (Fig. 1b). Note that for 10X Genomics data, we used the variant set discovered, genotyped, and phased by 10X' LongRanger software and hence variants not discovered decrease our estimate of completeness. The comparatively low percentage for Strand-seq can be explained by the relatively low sequencing coverage employed, combined with a slight unevenness in genomic coverage (Supplementary Fig. 2). For all technologies except Strand-seq, only short-range haplotypes were assembled using the read-based phasing, with a limited number of alleles phased per haplotype segment (Fig. 1c). For instance, we found >30,000 unconnected haplotype segments assembled from Illumina data, with the largest segment of 16 kb (median ~500 bp) harboring only 0.06% of the phased variants. This is because heterozygous variants that are further apart than the length of the sequenced DNA fragments cannot be connected, resulting in multiple disjoint haplotype segments with an unknown phase between them. Improvements were achieved using longer sequencing reads from PacBio technology, which effectively

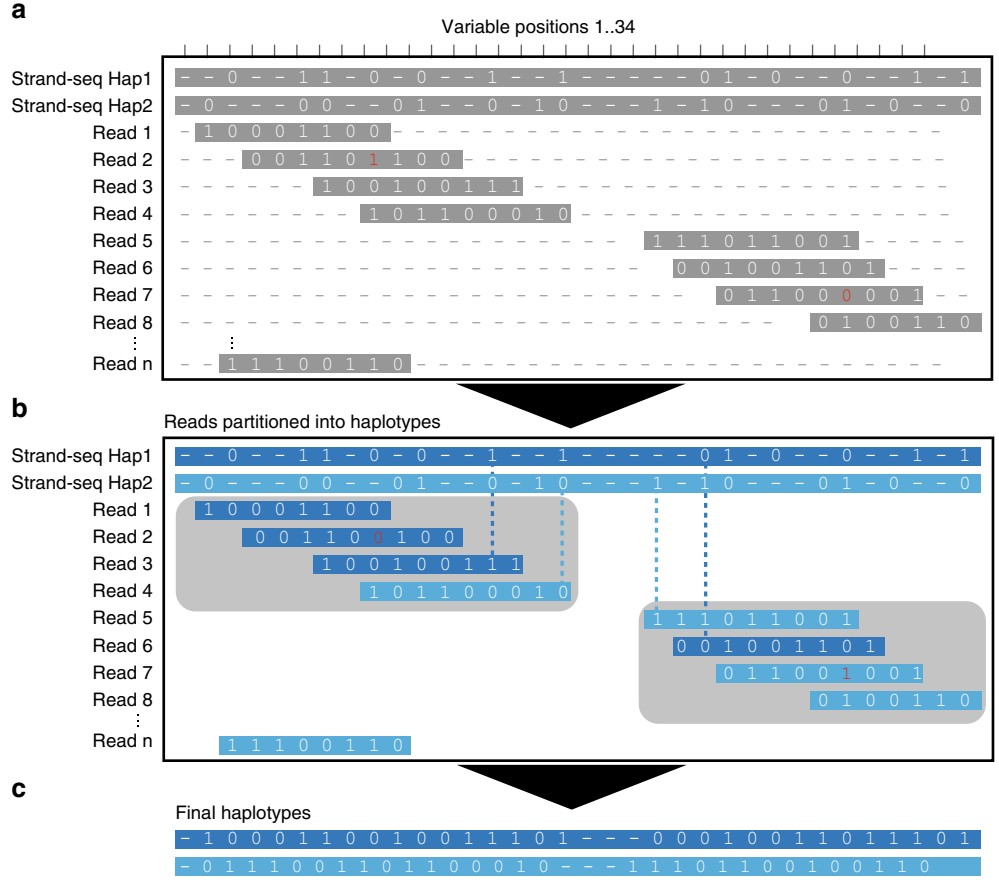

**Fig. 2** Integration of global and local haplotypes by the WhatsHap algorithm. An example solution of the weighted minimal error correction problem (wMEC) using WhatsHap algorithm is shown. For simplicity base qualities used as weights are omitted from the picture (for details on wMEC see Patterson et al. 2015). **a** The columns of the matrix represent 34 heterozygous variants (SNVs). Continuous stretches of zeros and ones indicate alleles supported by respective reads (0—reference allele, 1—alternative allele). The first two rows of the wMEC matrix are represented by Strand-seq haplotypes, illustrated as one "super read" connecting alleles along the whole length of the chromosome. (First row haplotype 1 alleles, second row haplotype 2 alleles). Subsequent rows of the matrix are represented by reads that map to the reference assembly in short overlapping segments. Sequencing errors (shown in red in read 2 and 7) are corrected when the cost for flipping the alleles is minimized. **b** Reads are then partitioned into two haplotype groups (Haplotpye 1— dark blue, Haplotype 2—light blue) such that a minimal number of alleles is corrected (in red). As an illustration of long haplotype contiguity facilitated by Strand-seq "super reads," we depict two non-overlapping groups of reads (gray rectangles) that can be stitched together by Strand-seq (dashed lines). **c** Final haplotypes are exported for both groups of optimally partitioned reads

decreased the number of phased haplotype segments (1927) and increased their size; the largest segment of 1.7 Mb (median ~21 kb) containing 1.25% of all SNVs on chromosome 1 (Fig. 1c). 10X Genomics produced even longer haplotype segments than both Illumina and PacBio data. The largest haplotype segment contained almost 5% of the heterozygous SNVs and spanned more than 8.5 Mb (median ~241 kb). Still, the haplotypes of chromosome 1 came in 199 disconnected segments and, hence, an end-to-end phasing was not achieved (Fig. 1c). That is, the linked reads from 10X Genomics were not able to connect distant heterozygous sites, for instance at centromeres, genome assembly gaps, or regions of low heterozygosity (Fig. 1a). This is in contrast to the global, albeit sparse, haplotypes produced by Strand-seq. Although the completeness of Strand-seq haplotypes was lower compared to the other technologies, all phased variants were placed into a single haplotype segment spanning the entire length of chromosome 1 (Fig. 1b, c).

Finally, we assessed the accuracy of each technology by calculating the extent of switch errors in comparison to the reference haplotypes. High-phasing accuracy of each technology was exemplified by the low percentage (<0.4%) of switch errors (Fig. 1d) with PacBio and 10X Genomics being the most accurate.

Since no single-phasing technology was sufficient to generate both global and dense haplotypes, we explored integrative phasing approaches that combine global, sparse haplotyping as afforded by Strand-seq technology with local high-density haplotypes from read-based phasing.

**Integrative global phasing strategy.** To generate more complete and dense haplotypes, we sought to establish a novel and integrative phasing approach using a combination of Strand-seq data with other data types. That is, we aim to enrich the sparse yet global phasing from Strand-seq using the dense haplotype information provided by Illumina, PacBio, or 10X Genomics. However, integrating phase information across platforms poses a non-trivial statistical and algorithmic challenge, which we resolved by treating the sparse Strand-seq haplotypes generated by StrandPhaseR as one row in the fragment matrix processed by WhatsHap (see "Methods" section). The other rows correspond to sequencing reads (PacBio, Illumina) or pre-assembled haplotype segments (10X Genomics) (see "Methods" section). This allows, for the first time, for integrative phasing by solving the corresponding optimization problem (weighted MEC) optimally (Fig. 2). We performed extensive experiments to demonstrate that

this approach enables excellent results in practice, as we describe in the following section.

To discover the most beneficial combinations of Strand-seq with Illumina or PacBio data, we explored combinations of variable numbers of Strand-seq libraries together with increasing depths of sequencing reads. To this end, we downsampled the number of Strand-seq libraries used in the analysis by randomly selecting subsets of libraries (5, 10, 20, 40, 60, 80, 100, or 120) from the original ($N = 134$) data set. Similarly, we randomly downsampled the sequencing reads from the Illumina and PacBio data sets to a lower genomic coverage (2, 3, 4, 5, 10, 15, 25, and 30-fold). We applied our integrative phasing strategy to all pairs

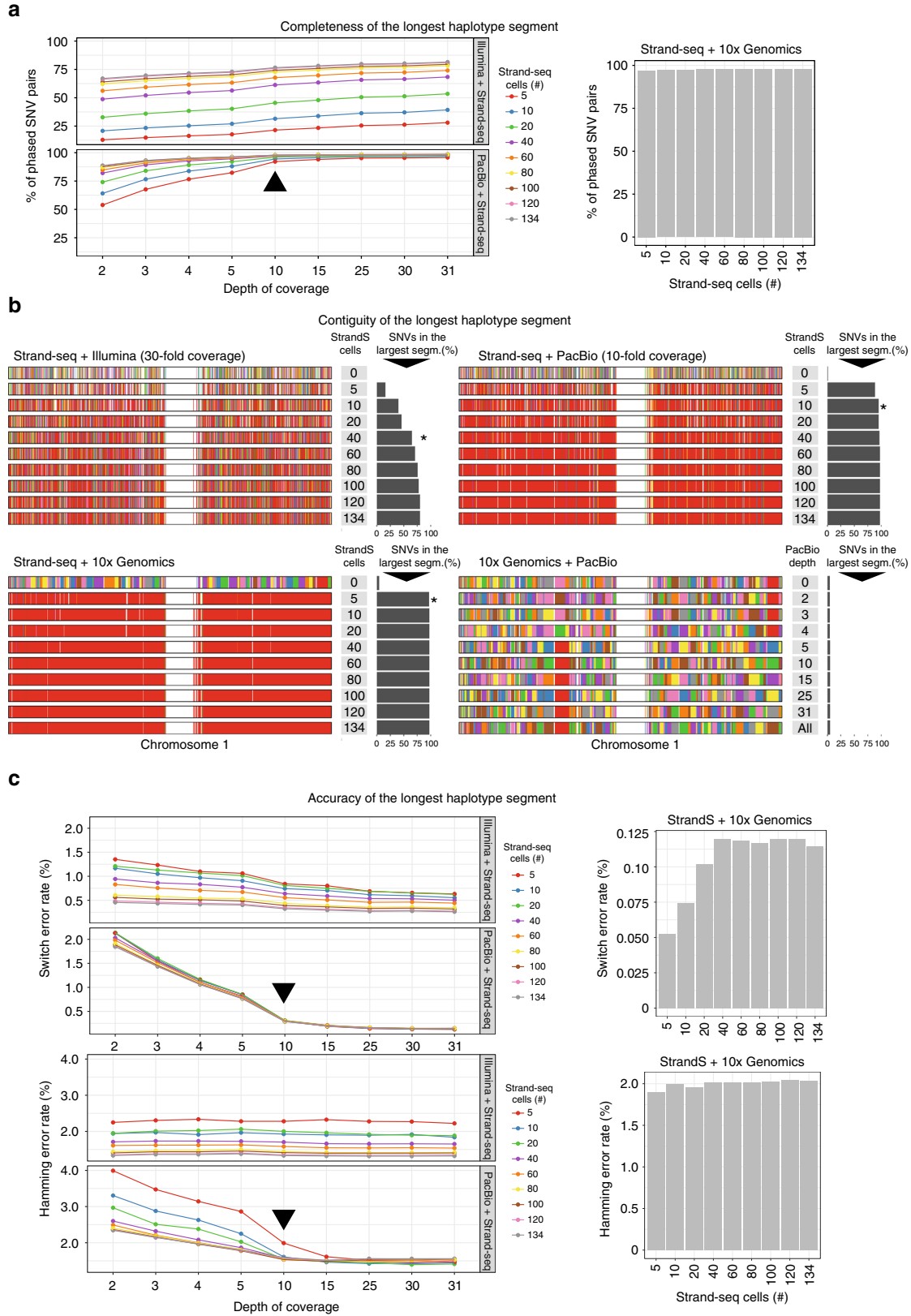

of downsampled Strand-seq libraries and the downsampled PacBio/Illumina data sets to assess the completeness (i.e., % of phased SNVs), contiguity (length of the largest haplotype segment), and accuracy (agreement with the "reference" standard) of each assembled haplotype.

We found that the combination of Strand-seq haplotypes with any of the other data types markedly increased the number of variants that were phased in the largest haplotype segment, albeit to differing degrees (Fig. 3a). Specifically, for the Illumina data, we observed the completeness of each haplotype increased gradually with the number of Strand-seq libraries used in the experiment, whereas the depth of coverage of Illumina data had only a minor but noticeable effect (Fig. 3a). In contrast, the PacBio data showed a significant improvement in haplotype completeness at 10-fold genomic coverage, regardless of the number of Strand-seq libraries used (Fig. 3a, black arrowhead). Similar results were seen when we combined Strand-seq with the 10X Genomics haplotypes (Fig. 3a). In all cases, integration of Strand-seq phasing drastically improved the contiguity of the haplotype spanning chromosome 1 (Fig. 3b). When combining Illumina data with 40 Strand-seq libraries, >65% of the reference variants could be phased accurately (Fig. 3b, black asterisk); 5497 haplotype segments (collectively representing 19.7% of the phased SNVs), however, remained disconnected, even when integrating the complete ($N = 134$) Strand-seq data set. These results confirm that Illumina data are of limited utility for haplotype phasing.

In contrast, as few as 10 Strand-seq cells combined with 10-fold PacBio coverage were sufficient to phase more than 95% of all heterozygous SNVs into a single haplotype segment (Fig. 3b, black asterisk), and merely five Strand-seq single-cell libraries were required to connect all 10X Genomics haplotypes. However, we recommend at least 10 Strand-seq libraries (Fig. 3b, black asterisk) to ensure that at least one haplotype-informative (i.e., Watson–Crick type) cell exists for every chromosome with high probability ($p = 0.978$). This global haplotyping was unique to Strand-seq, as the combination of 10X Genomics with PacBio reads proved inefficient to join locally phased segments (Fig. 3b). That is, the added value of combining these two technologies is limited as the haplotype segments tend to break at similar locations.

Finally, we assessed the phasing accuracy of the assembled haplotypes (the longest phased segment only) (Fig. 3c). Similar to the completeness of the haplotype, the accuracy of Illumina phasing gradually increased with sequencing depth and Strand-seq library number, indicating that Illumina coverage of 30-fold and higher is advisable (Fig. 3c). We further observed slightly elevated switch error rates at lower PacBio depths, which plateaued at 10-fold coverage (Fig. 3c, black arrowhead). This is likely caused by allele uncertainty resulting from error-prone PacBio reads, especially at lower sequencing depths. The lowest switch error rate (<0.2%) was achieved by the combination of Strand-seq with 10X Genomics data (Fig. 3c, right panel).

Switch error rates reflect local inaccuracies expressed by the number of pairs of consecutive heterozygous variants that are wrongly phased with respect to each other. These error rates are not necessarily informative about global haplotype accuracy,

which largely depend on how switch errors are spatially distributed (see "Methods" section and Supplementary Fig. 3a). Note that one single switch error implies that all following alleles (up to the next switch error) are assigned to the wrong haplotype. Since our goal is to generate dense and global haplotypes, we additionally report the Hamming error rate of the largest haplotype segment in comparison to the reference haplotypes (see "Methods" section and Supplementary Fig. 3b). Illumina reads are highly accurate and therefore we observed lower impact of sequencing depth on the global accuracy of the largest-phased haplotypes (Fig. 3c, Hamming error rate). In contrast, PacBio reads exhibited higher sequencing error rates, which translated into higher switch error rates at low sequencing depths. Using 10-fold PacBio coverage combined with at least 10 Strand-seq cells yielded highly accurate global haplotypes (Fig. 3c, black arrowhead), while lower coverages led to markedly worse results. Furthermore, the combination of Strand-seq with 10X Genomics haplotypes yielded highly accurate global haplotypes, already at the minimal amount of Strand-seq libraries (Fig. 3c, right panel).

Taken together, these results illustrate that Strand-seq can be used to phase existing sequence data and build dense, global and highly accurate haplotypes. Indeed, we found our approach highly efficient for genome-wide phasing (Fig. 4a). Using a combination of 40 Strand-seq libraries with 30-fold Illumina coverage, or 10 Strand-seq libraries with either 10-fold PacBio coverage or the 10X Genomics haplotypes, we successfully scaffolded chromosome-length haplotypes for every autosome of NA12878. The completeness of the genome-wide haplotypes measured for the largest haplotype segment reached 95.7 and 69.1% using PacBio and Illumina reads, respectively (Fig. 4a). We further demonstrated the high accuracy of these haplotypes on the local and global scales, which showed low switch (<0.45%) and Hamming error (<0.99%) rates for both the PacBio and Illumina combination (Fig. 4a). Whereas scaffolding the 10X Genomic haplotypes produced the most accurate local haplotypes (switch error rate of 0.05%), global performance suffered, and the highest Hamming error rate (2.18%) was calculated for this combination. Nevertheless, using Strand-seq to scaffold any of the data sets remarkably improved the completeness, contiguity, and accuracy of phasing for each chromosome, highlighting our integrative phasing strategy as a robust method for building dense and accurate whole-genome haplotypes (see "Data availability" to access phased SNVs for NA12878 using integrative phasing approach presented in this study).

## Discussion

Strand-seq has been successfully prepared from a wide range of cell types taken from various organisms[9,34,37] and is currently being adopted by an increasing number of researchers. The integrative phasing strategy we introduce here paves the way to leveraging Strand-seq to obtain chromosome-length, dense and accurate haplotypes at a manageable cost and labor investment. Based on the comprehensive evaluation presented above, we recommend three different combinations of Strand-seq with a complementary technology (Fig. 4b).

**Fig. 3** Various combinations of Strand-seq and read-based phasing using chromosome 1 as an example. Plots show haplotype quality measures for various combinations of Strand-seq cells (5, 10, 20, 40, 60, 80, 100, 120, 134) with selected coverage depths of Illumina or PacBio sequencing data (2, 3, 4, 5, 10, 15, 25, 30, >30-fold), or in combination with 10X Genomics haplotypes. **a** Assessment of the completeness of the largest haplotype segment as the % of phased SNVs. **b** Assessment of the contiguity of the largest haplotype segment as the length of the largest haplotype segment. Every phased haplotype segment is depicted as a different color, with the largest segment colored in red. Black asterisks point to a recommended depth of coverage of a given technology in combination with Strand-seq. **c** Assessment of the accuracy of the largest haplotype segment as the level of agreement with the "reference" standard. Black arrowheads highlight PacBio sequencing depth where accuracy of final haplotypes does not substantially improve.

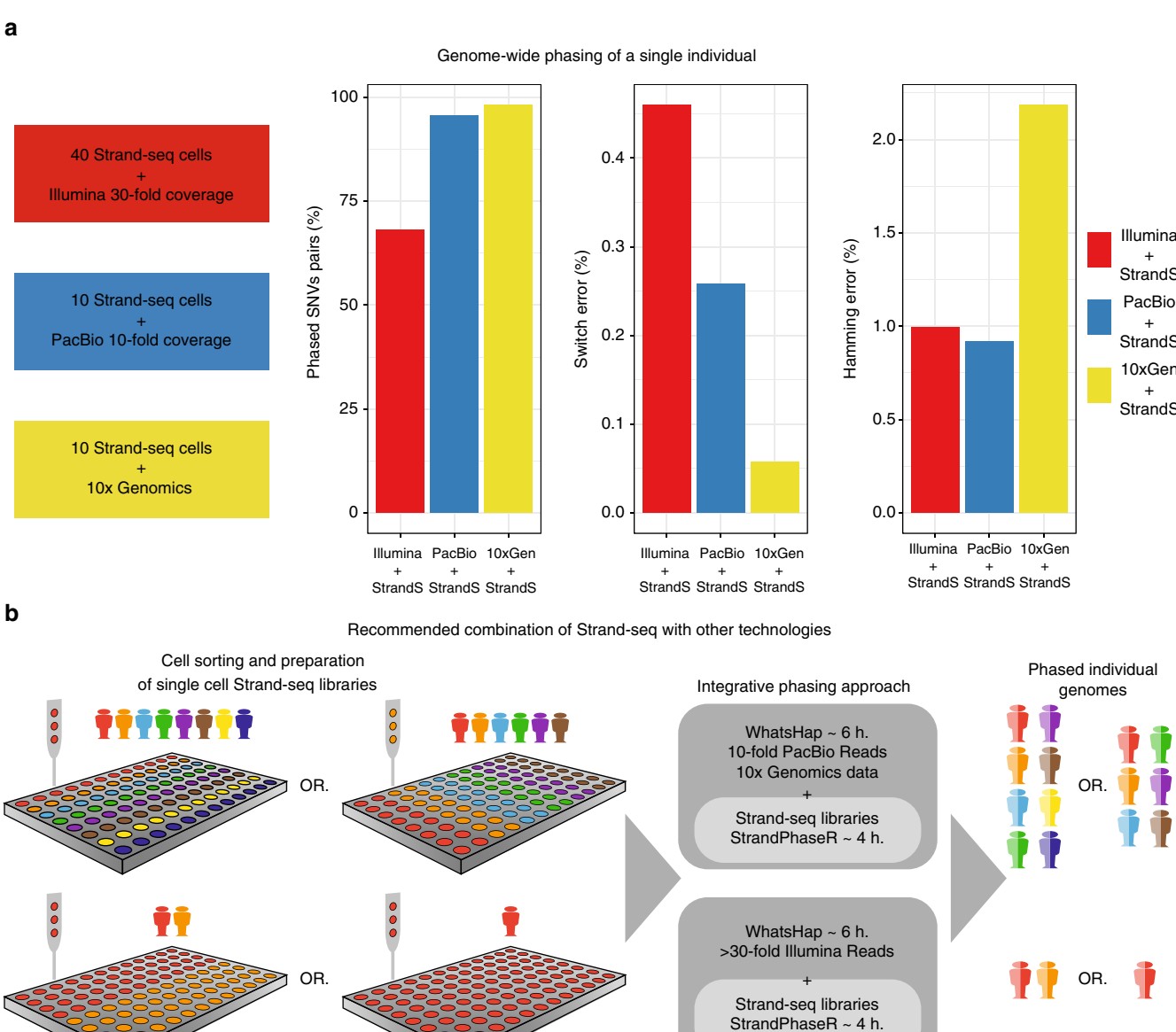

**Fig. 4** Recommended settings to phase certain amounts of individuals. **a** Genome-wide phasing of NA12878 using combination of 40 Strand-seq libraries with 30× short Illumina reads, 10 Strand-seq libraries with 10-fold long PacBio reads, or 10 Strand-seq libraries with 10X Genomics data. Plots show quality measures such as percentage of phased SNV pairs, switch error rate, and Hamming error rate for phased autosomal chromosomes. **b** A diagram providing the recommendations for the required number of Strand-seq libraries to be combined with recommended minimum of 10-fold PacBio and 30× Illumina coverage in order to reach global and accurate haplotypes for a depicted number of individual diploid genomes

As one option, one can combine Strand-seq with standard Illumina sequencing. Although the power of Illumina data for phasing is limited, mainly due to short insert sizes and read lengths, it still has some merit for adding variants to Strand-seq haplotypes. This might be of interest to many researchers since Illumina sequencing still constitutes the most common technology and there is an abundance of Illumina sequence data currently available for many sample genomes. To completely phase these pre-existing data, we recommend generating at least 40 Strand-seq libraries for the sample genome, which is sufficient to phase >68% of all heterozygous variants genome-wide with good accuracy (switch error 0.45%, Hamming error 0.99%), see Fig. 4a and Supplementary Table 3.

To build more complete haplotypes, we recommend combining Strand-seq with either PacBio or 10X Genomic technologies. A minimum of 10-fold PacBio coverage coupled with 10 Strand-

seq libraries will phase >95% of heterozygous variants genome-wide with excellent accuracy (switch error 0.25%, Hamming error 0.91%). Long-read sequencing has been demonstrated to be particularly powerful for resolving structural variation[38,39] and, although not explored here, might hence be the best choice when the resolution of haplotypes, structural variation, and repetitive regions is desired. However, the cost of this platform is still comparatively high. Therefore, until long-read technologies have become standard practice, we recommend combining 10 Strand-seq libraries with 10X Genomics technology. We found this combination yielded the most complete (>98% heterozygous variants genome-wide) haplotypes with the lowest switch error rate (0.05%). We did observe a slightly increased Hamming error rate (2.18%), however, which indicates that some genomic intervals are placed on the wrong haplotype, most likely due to switch errors in the pre-phased haplotype segments (produced by

10X Genomics) used as input. Overall, combining Strand-seq with 10X Genomics is the most cost-effective (in terms of time and money) strategy to phase an individual genome at extraordinary accuracy.

In this study, we used pre-phased 10X Genomics haplotype segments because using the raw sparse linked-read data leads to algorithmically challenging wMEC problem instances, which presently cannot be solved optimally by WhatsHap. This implies that variants that have not been discovered by LongRanger are considered unphased (and hence decrease "completeness") and that the error rates can likely be improved further by solving the combined instance resulting from Strand-seq and 10X Genomics data. We therefore consider processing the 10X Genomics raw data an important topic of future research.

In this paper, we focused on single individual haplotyping to avoid the biases and limitations of reference-panel-based phasing as well as the need to have access to genetic material of the parents, which might not be available in all settings, including clinical contexts. In cases when high-coverage sequencing data of the parents are available, such data sets can be used to enhance read-based phasing and provide long-range phase information[40] (Supplementary Note 3).

Strand-seq relies on BrdU incorporation during DNA replication and its use is therefore restricted to dividing cells. To provide long-range phase information for single samples in situations where growing cells are not available, Hi-C can constitute an alternative solution able to yield chromosome-spanning haplotypes[32]. However, the required coverage, and hence sequencing cost, is considerably higher for Hi-C than for Strand-seq. While the 134 Strand-seq libraries we used here reach a cumulative sequencing coverage of around 5×, markedly higher coverages are needed for Hi-C[32].

Our results demonstrate that dense and accurate chromosome-length haplotypes can be generated at manageable costs. This development brings haplotype-level analyses closer to a routine practice, which can be a key for understanding disease phenotypes. We emphasize that the strategy we present here works for single individuals without relying on other family members or statistical inference from haplotype reference panels. In contrast to such population-based phasing approaches, the method we advocate here allows insights into rare and de novo variants and long-range epistatic effects.

Our future efforts will focus on de novo assembly of haplotype-resolved genomes without the alignment to a reference genome. This will provide us with true diploid representations of individual genomes, which will have profound implications to study variability of personal genomes in health and disease.

## Methods

**Publicly available data sets used in this study**. Illumina reads[42,43] were obtained from 1000 Genome Project Consortium (http://ftp.1000genomes.ebi.ac.uk/vol1/ftp/phase3/data/NA12878/high_coverage_alignment/). PacBio reads[41] were downloaded from Genome in a Bottle Consortium (GIAB) (http://ftp.trace.ncbi.nlm.nih.gov/giab/ftp/data/NA12878/NA12878_PacBio_MtSinai/sorted_final_merged.bam). Pre-assembled 10X Genomics haplotypes (produced on the Chromium platform with Chromium Genome v1 reagents, sequenced on an Illumina HiSeq X Ten and processed with LongRanger 2.1.0) were downloaded from 10X Genomics website (https://support.10Xgenomics.com/genome-exome/datasets/NA12878_WGS_210) and filtered for heterozygous and PASS filter SNVs. Strand-seq libraries[9] have been downloaded from the European Nucleotide Archive (http://www.ebi.ac.uk/ena), accession number: PRJEB14185. Alternatively, aligned Strand-seq data used in this study can be obtained from Zenodo (doi:10.5281/zenodo.830278). As a gold standard, we downloaded pedigree-based haplotypes of NA12878 released as part of the Illumina platinum genomes (Version: 2016-1.0 from 6 June 2016) (http://www.illumina.com/platinumgenomes/).

**StrandPhaseR pipeline**. To build whole-genome haplotypes from Strand-seq data, we developed a new sorting-based pipeline, called StrandPhaseR. StrandPhaseR implements an improved phasing algorithm based on a binary sorting strategy of

two parallel matrices, storing haplotype information obtained from single-cell Strand-seq libraries. The analysis pipeline takes as input aligned BAM (binary alignment map) files from single cells, which were initially filtered for duplicate and low mapping quality reads (mapq <10). Haplotype-informative WC regions were localized in every Strand-seq library by counting the number of Crick (forward, "+") and Watson (reverse, "−") reads in equally sized regions (default 1 Mb). We used Fisher's exact test to calculate the probability that a region contained approximately equal numbers of Crick and Watson reads and agreed with the expected 50:50 ratio of a WC region[37]. Alleles at variable positions (supplied as a set of SNVs obtained from Illumina platinum haplotypes) were identified separately for W and C reads in every informative region to generate low-density single-cell haplotypes that are then sorted by the phasing algorithm. The partial single-cell haplotypes are used to fill two matrices, where rows represent cells and columns represent covered variable positions (SNVs) in any given cell (Supplementary Fig. 1). Initially, one matrix stores all variable positions found within the Watson templates, and a second matrix stores all variable positions found within the Crick templates. Cells in the matrices are sorted in decreasing order based on the number of covered variants (i.e., depth of coverage). Initially, a score of each column is calculated as the sum of all covered variants minus the most abundant variant. This represents the level of disagreement across all cells for the given SNV in the column. The sum of scores for each column represents the overall score of the matrix, and a lower matrix score represents a higher level of concordance across all SNV positions. Once the score of both matrices is determined, all SNVs in the first row (i.e., those belonging to the first cell) are swapped between the two matrices. In essence, this exchanges the Watson and Crick template strands of the cell within the matrix, to test whether there is a higher level of agreement across the phased SNVs found for all the cells. To determine this, the matrix scores are recalculated and if the scores are lower than the previous scores, the change is kept, otherwise the change is reversed. The algorithm continues with the second row. Again, the covered variants of the second cell are swapped between matrices, the matrix score is recalculated and the decision to preserve or reverse the change is made. This is repeated through all rows (cells) of the matrix, sorting the single-cell haplotypes within both matrices to reduce the number of conflicting alleles within each column. We repeated this sorting process twice, after which we did not observe any further changes. The resulting haplotypes are reported as the consensus allele found across all the cells for each column of the matrices. Ideally, there is only one allele present for every variable site in each matrix, however sporadic sequencing errors or cell-specific artefacts can introduce discrepancies. Lastly, any missing alleles at heterozygous sites are rescued by searching within the "uninformative" reads (i.e., those from WW and CC regions) present in Strand-seq libraries and filled in. The final consensus haplotypes are exported in a standardized VCF format, with each variable position that has an assigned Phred quality score and entropy value reflecting the confidence in the given allele. All phasing steps of StrandPhaseR have been implemented into a single open-source R package (see "Code availability").

**Downsampling of sequencing data**. To assess different combinations of Strand-seq libraries (w.r.t. number of single-cell libraries) with read data (w.r.t. depth of coverage), we performed a systematic analysis of the phasing performance for various subsets of each data set. To achieve this, we downsampled the original publicly available data sets consisting of 134 single-cell Strand-seq libraries[9], 45.8× coverage long-read PacBio data[41], and 41.1× coverage short-read Illumina data[42,43]. To simulate Strand-seq data sets consisting of reduced numbers of single cells, we randomly selected subsets of either 5, 10, 20, 40, 60, 80, 100, or 120 libraries from the original number of 134 libraries in the data set. Read data from the PacBio and Illumina data sets were downsampled using Picard (picard-tools-1.130) to meet a defined depth of coverage of either 2, 3, 5, 10, 15, 25, or 30-fold. The downsampling was performed for five independent trials to account for variability in downsampled data sets, and the average phasing performance across all trials was reported (as described below).

**Integrative phasing using WhatsHap**. As an input for integrative phasing, Strand-seq haplotypes were phased using StrandPhaseR (exported in VCF format) and combined with either PacBio or Illumina alignments (both stored in BAM format) or 10X Genomics pre-phased haplotype segments (stored in the VCF produced by LongRanger) to phase heterozygous variants obtained from Illumina platinum genomes. We achieved this integrative phasing across platforms by solving the weighted MEC (wMEC) problem using WhatsHap[19,20].

Mathematically, aligned reads from Illumina or PacBio (or pre-phased 10X Genomics haplotype segments) and sparse Strand-seq haplotypes are jointly represented in the form of a fragment matrix, where each row represent either one reads (in case of Illumina and PacBio), one pre-phased haplotype segment (in case of 10X Genomics), or one sparse global haplotype (in case of Strand-Seq data) and columns represent the variant sites (Fig. 2). The matrix is filled with 0, 1, and "−" entries, where 0 and 1 indicate that the corresponding read supports the reference or alternative allele, respectively, and "−" means the information is missing (e.g., because a read does not cover this variant site). WhatsHap selects a subset of rows and solves the wMEC problem optimally on these rows, as described earlier[20]. The result is a maximum likelihood bipartition of rows, which corresponds to the two sought haplotypes.

For all analyses, WhatsHap was provided with a reference genome (option "--reference") to enable re-alignment-based allele detection when constructing the fragment matrix from sequencing reads. This has been shown to significantly improve performance for PacBio reads[20].

**Quality metrics of assembled haplotypes**. To assess the quality of assembled haplotypes in this study, we calculated different metrics described in the following.

Completeness: The process of haplotyping establishes phase relations between pairs of consecutive heterozygous variants. We call each such pair a "phase connection." For each haplotype segment produced by a technology (or combination of technologies), we therefore count the number of phase connections, which is equal to the number of heterozygous markers that make part of such a haplotype segment minus one. To measure the completeness of a phasing, we sum the number of phase connections across all haplotype segments and divide by the maximum possible number of phase connections, which is equal to the number of heterozygous variants on a chromosome minus one.

Switch error rate: The switch error rate is the fraction of phase connections for which the phasing between the two involved heterozygous variants is wrong (Supplementary Fig. 3a).

Largest haplotype segment: In this study, we are interested in haplotypes that span the whole length of a chromosome. To measure the completeness of phasing, we report the fraction of heterozygous variants that are part of the largest haplotype segment.

Largest haplotype segment Hamming rate: To assess whether haplotypes are correct over long genomic distances, we only consider the largest haplotype segment and compute the Hamming distance between true and predicted haplotypes (Supplementary Fig. 3b), divided by the total number of heterozygous variants in this haplotype segment. That is, the Hamming error rate is equal to the fraction of wrongly phased heterozygous variants. Note that, only one switch error (e.g., in the middle of a chromosome) can result into a very high Hamming distance and hence the Hamming distance is a much more stringent quality measure. While the switch error rate assesses whether haplotypes are correct locally, i.e., between pairs of neighboring heterozygous variants, the Hamming distance assesses whether haplotypes are correct globally.

**Data availability**. We have made the final haplotypes resulting from using 134 Strand-seq cells and full PacBio coverage available via the GIAB FTP site (http://ftp-trace.ncbi.nlm.nih.gov/giab/ftp/data/NA12878/analysis/MPG_WhatsHap_phasing_07202017/).

**Code availability**. The StrandPhaseR software is publicly available through GitHub (https://github.com/daewoooo/StrandPhaseR). The WhatsHap software is publicly available through BitBucket (https://bitbucket.org/whatshap/whatshap). The computational pipeline to run experiments is available at (https://github.com/daewoooo/IntegrativePhasingPaper).

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

## Acknowledgements

D.P. was supported by an EMBO Short Term Fellowship (ASTF No: 624–2015). The Korbel group was supported by the National Human Genome Research Institute of the National Institutes of Health (NIH) under the award number U41HG007497. The content is solely the responsibility of the authors and does not necessarily represent the official views of the NIH. P.M.L. was supported by an Advanced ERC grant.

## Author contributions

D.P. and T.M. designed the study. D.P., S.G., and T.M. implemented the pipeline and performed experiments. D.P. prepared the figures. D.P., A.D.S., T.M., and S.G. wrote the manuscript, with the help of all authors. A.D.S., V.G., P.M.L., and J.O.K. helped with data interpretation. All authors read and approved final version of the manuscript.

## Additional information

**Competing interests:** The authors declare no competing financial interests.

