## [Peer Review File · Nature Communications]

Reviewers' comments:

Reviewer #1 (Remarks to the Author):

The authors present novel methods that combine their previously published read-based phasing methods with the previously published strandseq method. These methods allow them to achieve relatively accurate and complete global phasing that is not possible with either method alone. The authors do a very good job evaluating different combinations of technologies and coverage to guide the user in selecting technologies for their use case. I expect this method will be quite useful to the community, particularly if strandseq is streamlined and made less labor intensive over time. I recommend publication of this work after the points below are addressed:

When the authors calculate the completeness, is the denominator always all heterozygous SNPs in the platinum genomes vcf? If so, presumably any variants missing or called homozygous in the 10X vcf would count against completeness? This is fine, but it is useful to explain this clearly. It's also worth noting that platinum genomes is not completely comprehensive in characterize variants, particularly in difficult regions, so "completeness" metrics would not account for variants in very difficult regions.

Were indels included in the authors' definition of SNVs?

Methods part 2: "to assess difference" should be "to assess different"

The mean pacbio read length reported here is much longer than in the original paper - could the authors verify that this is correct? The original paper in the supplementary methods says "Sequencing of NA12878 genomic DNA was conducted across 851 Pre P5-C3 (predominantly XL-C2) and 162 P5-C3 SMRTcells to generate 24X and 22X coverage with aligned mean read lengths of 2425 (2993, raw subread) and 4891 (5884, raw subread) base pairs, respectively." - <http://www.nature.com/nmeth/journal/v12/n8/abs/nmeth.3454.html#supplementary-information>

What was the 10X coverage, chemistry version, and version of analysis pipeline?

Which version of platinum genomes calls were used?

Fig 3c(II) - why switch error of zero with no strandseq?

Fig 3c(iv) - I recommend the authors don't show hamming error at 0 since it is uninformative and makes it difficult to see smaller values comparable to 3c(iii)

I really like how the authors calculate both switch error and hamming error rates. I think it could also be useful to calculate the hamming error rate in a slightly different way for the methods that do not have global phasing. In particular, could the authors calculate hamming error in each locally phased segment vs. the benchmark, and aggregate this hamming error across all locally phased segments? I understand this would be different from the global hamming error, but I think it would be a different useful statistic. This could be a useful feature to add to the authors' whatshap tool

Could the authors compare and contrast the results in this paper with the trio+reads-based method they published previously?

Could the authors make the resulting phased vcfs available? The Genome in a Bottle ftp site may be able to host these, and I'd be happy to help with this.

Could the authors give an example of the command line used to phase with both strandseq and pacbio or 10X?

Justin Zook
National Institute of Standards and Technology

Reviewer #2 (Remarks to the Author):

In this manuscript, the authors proposed a "hybrid" haplotyping method for generating dense and chromosome-length haplotypes by integrating Strand-seq with PacBio or 10X Genomics. Using the NA12878 platinum genome as the reference, they evaluated multiple combination of methods and data sizes in terms of accuracy and fractions of variant phased, and recommended optimal method combinations for routinely generation of fully phased diploid genome.

Overall the study was carefully designed, the analysis was performed rigorously, and the conclusions are sound and informative to the community.

I do have one major and one minor criticism:

(1) The authors published a similar paper on Genome Research last year "DNA template strand sequencing of single-cells maps genomic rearrangements at high resolution". It is not entirely clear to me what are the major advances of this manuscript over the 2016 ones. How many of the 143 Strand-seq data sets were generated in that previous study? Is the StrandPhaseR software a major upgrade over the StrandPhase software developed previously? The previous study also include PacBio data in the analysis, and the inclusion of 10X Genomics data appears to be new. I feel that the authors need to be more explicit on what is new compared with their previous work, which is critical for the assessment of the significance of this manuscript.

(2) It would be helpful to provide some cost estimates of the haplotyping solutions they recommend to the community, perhaps by adding one more column to Table S2.

(3) I found the part of introduction related to connecting enhancers and target genes misleading and unnecessary. Fully phased haplotypes do not really solve that problem, a high-density HiC map does.

Reviewer #3 (Remarks to the Author):

The authors present the use of Strand-seq libraries as a source of haplotype-specific sequence information that can be used for long range scaffolding of more dense haplotyping experimental tools, of which the authors explore several. Overall the manuscript is very well written, has all necessary details, and is very clear in its presentation of the results. I believe the manuscript is suitable for publication in Nature Communications with only minor revisions.

In general, the use of Strand-seq data for haplotype resolution is a great idea. When the method initially came out I was very excited about this potential application of the technology, but did not have bandwidth to explore the concept. The authors of this manuscript not only explore it thoroughly, but provide software to carry out the analysis and explore the level of sequencing and libraries required to perform high-quality phasing.

The primary advantage of Strand-seq for phasing is that it allows for the spanning of large repetitive

or homozygous regions that cannot be traversed by lower-contiguity methods. Currently, Hi-C based strategies can also provide this form of information (as the authors clearly state and cite), however the cost of additional sequencing of the Hi-C library is likely far greater than for Strand-seq. On this point - the authors could improve this argument by putting some numbers behind the cost differences.

One comment regarding the haplotype resolution using 10X data: the authors use phased blocks that are the output of the longranger 10X pipeline, but also note that this results in a fairly high number of chimeric haplotype blocks. To get around this, it seems the best strategy may be to instead use the SNV information from read clouds as opposed to relying on the 10X software. This would likely slow down the phasing using the authors tool, but I anticipate would produce better results overall. That being said - I am not sure how open the read cloud information is from the 10X platform, but the authors should certainly attempt this approach.

The authors use the term "Illumina sequencing" when referring to shotgun sequencing libraries that are sequenced on the Illumina platform. This should instead be referred to as "short fragment sequencing", "shotgun sequencing", "Illumina shotgun sequencing", or something similar to make it clear that it is shotgun sequencing. For instance, 10X genomics uses Illumina sequencing, as does Hi-C, Strand-seq, etc...

The authors carry out their analysis using only the platinum quality phased SNVs. Many genomes would not have such a highly curated high quality set of SNV calls. Performing SNV calling and then phasing could be of value as an intermediate to the goal stated in the discussion to perform de novo assembly and phasing.

Lastly, the most important criticism I have is that Strand-seq libraries require the culturing of cells to incorporate U's on one of the strands to be ablated / degraded in order to achieve the strand-specific information. This may not be possible for a number of samples, whereas something like Hi-C could be (e.g. a frozen biopsy etc...). While this is not a deal-breaker for the value of the technology, it is an important limitation that the authors need to state and address.

Dear Editor,

thank you for the positive initial assessment of our manuscript “Dense and accurate whole-chromosome haplotyping of individual genomes”. We have revised the manuscript and supplementary materials to address all the points raised by the reviewers. We provide detailed point-by-point responses to each of the reviewers’ comments below.

Reviewer #1

Comment: “The authors present novel methods that combine their previously published read-based phasing methods with the previously published strandseq method. These methods allow them to achieve relatively accurate and complete global phasing that is not possible with either method alone. The authors do a very good job evaluating different combinations of technologies and coverage to guide the user in selecting technologies for their use case. I expect this method will be quite useful to the community, particularly if strandseq is streamlined and made less labor intensive over time. I recommend publication of this work after the points below are addressed:”

Response: We thank the reviewer this positive assessment. In the meantime, a detailed protocol description for Strand-seq has appeared (Sanders et al., Nature Protocols, 12:6, 1151-1176, May 2017), which will help a wider adoption of the Strand-seq method. Given the current advances in single-cell technology, we are confident that the protocol can be scaled up substantially in the near future, making it more attractive also for larger cohorts. Note that the required Strand-seq coverage per sample is very low (see below).

Comment: “When the authors calculate the completeness, is the denominator always all heterozygous SNPs in the platinum genomes vcf? If so, presumably any variants missing or called homozygous in the 10X vcf would count against completeness? This is fine, but it is useful to explain this clearly. It’s also worth noting that platinum genomes is not completely comprehensive in characterize variants, particularly in difficult regions, so “completeness” metrics would not account for variants in very difficult regions.”

Response: We agree that the platinum genomes we use for comparison are almost surely incomplete in difficult genomic regions such as segmental duplications, which we now mention in the Results section. We chose to use this call set because the large pedigree that was sequenced leads to extremely accurate haplotypes, which is instrumental for estimating phasing error rates. For the reader’s reference we now additionally include a comparison of sites and genotypes to the latest Genome in a Bottle release (v.3.3.2) call set (**Supplemental Note 1**). The Platinum Genomes SNV set is larger than the GIAB set (3,523,429 vs. 3,257,172 bi-allelic SNVs). The genotype concordance on the intersection is almost perfect (99.9998%).

Comment: “Were indels included in the authors’ definition of SNVs?”

Response: No, in our definition of SNVs we do not include indels. However, they can straightforwardly be included during the integrative WhatsHap phasing runs. We have now

done this, and provide corresponding results for all combinations of Strand-Seq libraries with various depths of PacBio reads (**Supplemental Note 4** and **Supplemental Fig. S6**). Note that the phasing error rates for indels are much higher than for SNVs: when combining the full Strand-seq data set with full PacBio coverage, we observe 18.4% additional switch errors (extra switches divided by extra phase connections, see caption of **Supplemental Fig. S6**). This is consistent with our earlier findings (Martin et al., biorxiv, 2016, doi: 10.1101/085050, Figure 6), where we obtained similar error rates of WhatsHap (and much worse rates for other tools). High error rates when phasing indels are likely due to difficulties in sequencing, aligning and assembling indels, particularly such in low-complexity DNA (STRs, VNTRs, etc.).

Comment: Methods part 2: “to assess difference” should be “to assess different”

Response: Fixed.

Comment: “The mean pacbio read length reported here is much longer than in the original paper - could the authors verify that this is correct? The original paper in the supplementary methods says “Sequencing of NA12878 genomic DNA was conducted across 851 Pre P5-C3 (predominantly XL-C2) and 162 P5-C3 SMRTcells to generate 24X and 22X coverage with aligned mean read lengths of 2425 (2993, raw subread) and 4891 (5884, raw subread) base pairs, respectively.”

Response: We thank the reviewer for spotting this, and apologize for this mistake. We recomputed coverages and corrected the number to 4429 bp, which is the average length of reads with a primary alignment to Chromosome 1 (**Supplemental Table S1**).

Comment: “What was the 10X coverage, chemistry version, and version of analysis pipeline?”

Response: According to 10X Genomics, the data set was created on a Chromium instrument using Chromium Genome v1 reagents, sequenced on an Illumina HiSeq X Ten and processed with LongRanger 2.1.0. This information has been added to the manuscript (see **Data Access**).

Comment: “Which version of platinum genomes calls were used?”

Response: Version 2016-1.0 from 6 June 2016 for hg19 was used in this study. We have added this information to the manuscript (see **Data Access**).

Comment: “Fig 3c(ii) - why switch error of zero with no strandseq?”

Response: Panel C of Figure 3 reports statistics on the largest haplotype segment. In this case, the largest segment indeed did not have a single switch error. Note that the largest segment is quite short unless Strand-seq data are being used. We have removed this entry (just as in 3c(iv), see next comment below) to avoid confusion about the drastically different denominator compared to all entries with >0 Strand-seq cells (which places most variants into the largest block).

Comment: “Fig 3c(iv) - I recommend the authors don't show hamming error at 0 since it is uninformative and makes it difficult to see smaller values comparable to 3c(iii)”

Response: We agree and we have removed that data point from the plot.

Comment: “I really like how the authors calculate both switch error and hamming error rates. I think it could also be useful to calculate the hamming error rate in a slightly different way for the methods that do not have global phasing. In particular, could the authors calculate hamming error in each locally phased segment vs. the benchmark, and aggregate this hamming error across all locally phased segments? I understand this would be different from the global hamming error, but I think it would be a different useful statistic. This could be a useful feature to add to the authors’ whatshap tool”

Response: We thank the reviewer for this very good suggestion, and agree that this additional quality measure is helpful. We have added this feature to WhatsHap’s compare subcommand (see commit 6070533, released in WhatsHap version 0.14.1) and now refer to this metric as “block-wise Hamming distance”. We have computed it for all compared data sets and include these results in **Supplemental Fig. S5**.

Comment: “Could the authors compare and contrast the results in this paper with the trio+reads-based method they published previously?”

Response: We agree that the proposed analysis can be very informative. Although the present paper focuses on single individual haplotyping, it is interesting to compare it with haplotyping based on trio data. To address this, we performed a phasing experiment where we use PacBio sequencing data of NA12878 in conjunction with genotype data of the parents, but no Strand-seq data. We downloaded the Illumina paired-end BAM files for NA12878 and her parents (platinum genome data), called and genotyped SNVs using FreeBayes for all three samples. We then performed phasing using WhatsHap, once in “pure trio mode” (i.e. only using family genotypes and no reads) and once by additionally providing PacBio reads of the child. In pure trio mode, all SNVs that are not “all het” (i.e. homozygous in at least one individual) are phased into one single block. For Chromosome 1, this approach reaches a completeness of 83.0% (fraction of phased heterozygous SNVs in the child) and a switch error rate of 0.024% (Hamming rate: 0.013%) when compared to the platinum genome VCF. The low error rates are not surprising given that the two phasings are based on the same data set. When additionally using PacBio data for the child, we reach a completeness of 96.1% (largest block) and a corresponding switch error rate of 0.12% (Hamming rate 0.11%). This shows that sequencing data of the parents can indeed be used in lieu of Strand-seq data. These results are now summarized as Supplementary Note 3 and referenced in the Discussion. However, we would like to highlight that parent samples are not available in all contexts (e.g. clinical) and that both parents need to be sequenced to sufficient depth so as to allow for reliable genotyping and phasing (which amounts to substantially more sequencing depth required for Strand-seq).

Comment: “Could the authors make the resulting phased vcfs available? The Genome in a Bottle ftp site may be able to host these, and I'd be happy to help with this.”

Response: We are very pleased by the reviewer's interest to make our long-range haplotype available through GIAB’s FTP site. We agree that this is a great way of sharing the data with the community and have uploaded our final phased VCF. It is now available from ftp://ftp-trace.ncbi.nlm.nih.gov/giab/ftp/data/NA12878/analysis/MPG_WhatsHap_phasing_07202017.

Comment: “Could the authors give an example of the command line used to phase with both strandseq and pacbio or 10X?”

Response: To ensure full reproducibility, we have made our Snakemake workflow available on github (<https://github.com/daewoooo/IntegrativePhasingPaper>). For convenience, we additionally describe the used WhatsHap command lines in **Supplemental Note 2**.

Reviewer #2

Comment: “In this manuscript, the authors proposed a "hybrid" haplotyping method for generating dense and chromosome-length haplotypes by integrating Strand-seq with PacBio or 10X Genomics. Using the NA12878 platinum genome as the reference, they evaluated multiple combination of methods and data sizes in terms of accuracy and fractions of variant phased, and recommended optimal method combinations for routinely generation of fully phased diploid genome.

Overall the study was carefully designed, the analysis was performed rigorously, and the conclusions are sound and informative to the community.”

Response: We thank the reviewer for these positive remarks.

Comment: “(1) The authors published a similar paper on Genome Research last year "DNA template strand sequencing of single-cells maps genomic rearrangements at high resolution". It is not entirely clear to me what are the major advances of this manuscript over the 2016 ones.”

Response: In our previous paper (Porubsky et al., Direct chromosome-length haplotyping by single-cell sequencing, Genome Research 26:1, 1-10, 2016), we pioneered the use of Strand-seq for haplotype phasing. This led to chromosome-length yet sparse haplotype scaffolds with a completeness of 57.6% of heterozygous SNVs being phased (using the libraries included in this study, also see next comment). By comparison, in the present paper we introduce and evaluate a novel computational strategy for integrative phasing using Strand-seq and a complementary data source (e.g. PacBio, 10X Genomics, or Illumina paired-ends). Importantly, this method allows to phase >98% of all heterozygous variants, hence representing a major advance over previously published results. To achieve this, we designed a novel framework for integrating diverse data types to generate highly accurate and complete haplotypes at chromosome-length scale. To evaluate this framework, we used publicly available data sets for NA12878, including the Strand-seq data from the previous Genome Research paper; however, moving forward we envision our model can be immediately implemented to phase existing sequencing data currently available for other genomes, and can serve as an important new guideline to inform future project designs. We have added a sentence to the Introduction to make the advances over the previous paper more explicit.

Comment: “How many of the 134 Strand-seq data sets were generated in that previous study?”

Response: All 134 Strand-seq libraries were produced as a part of the previous study. (We selected all libraries prepared with the paired-end protocol from the original dataset.)

Comment: “Is the StrandPhaseR software a major upgrade over the StrandPhase software developed previously?”

Response: We consider StrandPhaseR (implemented in R) as an important upgrade over the previous phasing pipeline (implemented in PERL), especially with respect to user-friendliness. StrandPhaseR is available as an R package, what makes use of this package easier to adopt by potential Strand-seq users. In addition, StrandPhaseR exports phased haplotypes in VCF format as well as splits reads from all cells into two haplotypes per chromosomes. We have added a corresponding statement to the Results section.

Comment: “The previous study also include PacBio data in the analysis, and the inclusion of 10X Genomics data appears to be new. I feel that the authors need to be more explicit on what is new compared with their previous work, which is critical for the assessment of the significance of this manuscript.”

Response: For clarification, in the previous study (Porubsky et al., GR, 2016) we have used PacBio *RNA-seq* data as a means to validate the correctness of inferred haplotypes. Indeed, no PacBio *DNA* sequencing reads were used, and further, no PacBio reads (neither DNA nor RNA) reads were *integrated* with Strand-seq to increase the coverage and/or precision of computed haplotypes. As stated above, the use of Strand-seq data in an integrated fashion with any other data type (10X, PacBio, Illumina paired-end based DNA sequencing data), with the objective to obtain more dense haplotypes, is a major novelty of our present manuscript. We have added a sentence to the Introduction section to clarify this.

Comment: “(2) It would be helpful to provide some cost estimates of the haplotyping solutions they recommend to the community, perhaps by adding one more column to Table S2.”

Response: We appreciate how a table with breakdown of costs would be helpful to inform experimental design, and thank the reviewer for this suggestion. However, the cost of library preparation and sequencing varies considerably depending on the platform used (e.g. MiSeq, HiSeq or specialized sequencing service), local, national or institutional reagent rates available to the researcher, and access to special discounts from suppliers. Moreover, the pace of technology developments is likely to make any cost estimate we provide quickly out-of-date. In light of this, we feel a more robust measure serving the community over longer terms is to report the minimal sequence depth requirements for each haplotyping solution. In response to this reviewer comment, we thus now include required coverages in **Supplemental Table S3**. We have also included a new Supplemental Figure that illustrates the depth of coverage obtained from the downsampled Strand-seq cells used in this study (**Supplemental Fig. S4**).

Comment: “(3) I found the part of introduction related to connecting enhancers and target genes misleading and unnecessary. Fully phased haplotypes do not really solve that problem, a high-density HiC map does.”

Response: We thank the reviewer for this comment. Indeed, our sentence on “interactions between enhancers and their target genes” was misleading. We re-phrased it (and the following sentence) and now more specifically speak of “epistatic interactions between variants in enhancers and variants in their target genes or their promoters”. This should remove the ambiguity between “interactions” in the sense of 3D contact and “interactions” in the sense of epistatic interaction.

Reviewer #3

Comment: “The authors present the use of Strand-seq libraries as a source of haplotype-specific sequence information that can be used for long range scaffolding of more dense haplotyping experimental tools, of which the authors explore several. Overall the manuscript is very well written, has all necessary details, and is very clear in its presentation of the results. I believe the manuscript is suitable for publication in Nature Communications with only minor revisions.

In general, the use of Strand-seq data for haplotype resolution is a great idea. When the method initially came out I was very excited about this potential application of the technology, but did not have bandwidth to explore the concept. The authors of this manuscript not only explore it thoroughly, but provide software to carry out the analysis and explore the level of sequencing and libraries required to perform high-quality phasing.”

Response: We thank the reviewer for being so positive about our study.

Comment: “The primary advantage of Strand-seq for phasing is that it allows for the spanning of large repetitive or homozygous regions that cannot be traversed by lower-contiguity methods. Currently, Hi-C based strategies can also provide this form of information (as the authors clearly state and cite), however the cost of additional sequencing of the Hi-C library is likely far greater than for Strand-seq. On this point - the authors could improve this argument by putting some numbers behind the cost differences.”

Response: We have informed ourselves about the costs of library preparation and learned that the costs of preparing a Hi-C library is comparable to the costs of a Strand-seq experiment corresponding to a 96-well plate (enabling the analysis 96 individual cells). Please note also that costs per library tend to fluctuate slightly from laboratory to laboratory, for example, depending on local product rates and rebates etc. In the light of this, we investigated the needs of each technology in terms of total DNA sequencing coverage, a value that is less volatile in terms of costs and hence can serve as an objective measure for the efforts required to pursue phasing using Strand-Seq versus Hi-C. We used the data from our Figure 1, which compares the number of SNVs covered in the largest haplotype block (which in the case of Strand-Seq is typically the entire chromosome) and the switch error rate of the resulting haplotype against similar estimates derived from the recent HapCUT2 paper (Edge et al. Genome Res. 2017) – a study that used Hi-C data from the NA12878 cell line to generate global haplotypes. According to Figure 4 from the HapCUT2 paper, we estimate that a coverage of ~40x in Hi-C data is required to achieve the density (‘fraction of SNVs in largest block’) and accuracy (‘switch + mismatch error rate for largest block’) seen from 134 Strand-Seq libraries. As illustrated in the new Supplemental Figure (**Supplemental Fig. S4**) the latter

amounts to ~5x genome sequencing coverage, which indicates that, when using Strand-Seq, similar quality haplotypes can be generated using approximately 8-fold reduced coverage (translating to required sequencing costs) compared to Hi-C. We now point out the increased coverage requirements of Hi-C compared to Strand-seq in the Discussion section.

Comment: “One comment regarding the haplotype resolution using 10X data: the authors use phased blocks that are the output of the longranger 10X pipeline, but also note that this results in a fairly high number of chimeric haplotype blocks. To get around this, it seems the best strategy may be to instead use the SNV information from read clouds as opposed to relying on the 10X software. This would likely slow down the phasing using the authors tool, but I anticipate would produce better results overall. That being said - I am not sure how open the read cloud information is from the 10X platform, but the authors should certainly attempt this approach.”

Response: The reviewer makes a s very good point – we agree that using the raw 10X read cloud data would be preferable. To this end, we have created an adapted version of WhatsHap (see “10xG_phasing” branch in WhatsHap git repository) able to read barcode information from 10X BAM files (encoded in BX tags). We followed the same procedure to select read clouds as in Zheng et al. (2016) such that reads within the read cloud span at most 50kb and have a mapping quality of at least 60. Such unique read clouds were then used as an input for WhatsHap pipeline, i.e. each read cloud is used as one input row in the fragment matrix. Note however that the WhatsHap pipeline reduces the maximum (physical) coverage of the input data to a manageable quantity (default 15x) before solving the NP-hard wMEC problem, since the runtime increases exponentially with this parameter. While only negligibly affecting results for regular (e.g. PacBio) sequencing reads, this approach is not ideal for sparse read cloud data. In our experiments (on chr1 of NA12878), only 34.4% of all read clouds could be retained and we observed a switch error rate of 0.43%, while the phasing provided by 10X Genomics (LongRanger) is more accurate (0.025% switch errors). Hence, we decided to not include these results from using 10X raw data here, but instead point to the open algorithmic challenge of integratively phasing from Strand-seq and 10X raw data in the Discussion section.

Comment: “The authors use the term "Illumina sequencing" when referring to shotgun sequencing libraries that are sequenced on the Illumina platform. This should instead be referred to as "short fragment sequencing", "shotgun sequencing", "Illumina shotgun sequencing", or something similar to make it clear that it is shotgun sequencing. For instance, 10X genomics uses Illumina sequencing, as does Hi-C, Strand-seq, etc...”

Response: We thank the reviewer for raising this point that might lead to confusion among the readers since now Illumina sequencing is heavily employed in producing read-clouds using 10X Genomic technology. We made clear that term ‘Illumina’ refer to only short-read Illumina sequencing in this study. We have added this statement at the beginning of the result section.

Comment: “The authors carry out their analysis using only the platinum quality phased SNVs. Many genomes would not have such a highly curated high quality set of SNV calls. Performing SNV calling and then phasing could be of value as an intermediate to the goal stated in the discussion to perform de novo assembly and phasing.”

Response: We thank the reviewer for this suggestion. We have performed SNV calling using FreeBayes, compared the resulting SNV call set to the GIAB and Platinum Genomes call sets, and observed very good concordance (see Supplementary Note 1 and Supplemental Fig. S8). We ran our phasing pipeline on the FreeBayes SNV call set and observed only marginally worse results than when starting from the Platinum call set (see Supplementary Note 1 and Supplemental Table S4). When combining PacBio (10-fold) with Strand-seq data (10 cells), for instance, the switch error rate increases from 0.25% to 0.34% and the Hamming error rate increases from 0.91% to 1.14%. We therefore conclude that our approach works almost equally well on an imperfect SNV call set resulting from a standard pipeline. In particular, we note that ability to correctly phase across whole chromosomes remains unaffected, as reflected in the low Hamming error rate.

Comment: “Lastly, the most important criticism I have is that Strand-seq libraries require the culturing of cells to incorporate U's on one of the strands to be ablated / degraded in order to achieve the strand-specific information. This may not be possible for a number of samples, whereas something like Hi-C could be (e.g. a frozen biopsy etc...). While this is not a deal-breaker for the value of the technology, it is an important limitation that the authors need to state and address.”

Response: We agree with this suggestion and have added a statement about this remaining limitation of Strand-seq to the Discussion section, and now mention Hi-C as an appropriate alternative for such scenarios.

REFERENCES

Edge, P., Bafna, V. & Bansal, V. HapCUT2: robust and accurate haplotype assembly for diverse sequencing technologies. *Genome Res.* 1–23 (2016).

Sanders, A. D., Falconer, E., Hills, M., Spierings, D. C. J. & Lansdorp, P. M. Single-cell template strand sequencing by Strand-seq enables the characterization of individual homologs. *Nat. Protoc.* 12, 1151–1176 (2017).

Zheng, G. X. Y. et al. Haplotyping germline and cancer genomes with high-throughput linked-read sequencing. *Nat. Biotechnol.* **34**: 303–311 (2016).

REVIEWERS' COMMENTS:

Reviewer #1 (Remarks to the Author):

The authors have done an excellent job responding to reviewer comments, and I recommend this for publication.

Reviewer #2 (Remarks to the Author):

The authors have addressed all the issues that I raised, and I recommend the publication of this manuscript on Nature Communications.

Reviewer #3 (Remarks to the Author):

The authors have thoroughly addressed all of the points I raised and I believe the revised manuscript is suitable for publication.